# Improving Exploration of Deep Reinforcement Learning using Planning for Policy Search

## Abstract

Most Deep Reinforcement Learning methods perform local search and therefore are prone to get stuck on non-optimal solutions. Furthermore, in simulation based training, such as domain-randomized simulation training, the availability of a simulation model is not exploited, which potentially decreases efficiency. To overcome issues of local search and exploit access to simulation models, we propose the use of kinodynamic planning methods as part of a model-based reinforcement learning method and to learn in an off-policy fashion from solved planning instances. We show that, even on a simple toy domain, D-RL methods (DDPG, PPO, SAC) are not immune to local optima and require additional exploration mechanisms. We show that our planning method exhibits a better state space coverage, collects data that allows for better policies than D-RL methods without additional exploration mechanisms and that starting from the planner data and performing additional training results in as good as or better policies than vanilla D-RL methods, while also creating data that is more fit for re-use in modified tasks.

## 1 Introduction

Robots in human-centric environments are confronted with less structured, more varied and more quickly changing situations than in typical automated manufacturing environments. Research in autonomous robots adresses these challenges using modern machine learning methods. However, learning and trying out actions directly on a real robot is time-consuming and potentially dangerous to the environment as well as to the robot. In contrast, physically-based simulation provides the benefit of faster, cheaper, and safer ways for robot learning.

If simulation models are available, they can be used by sampling-based planning methods that are able to directly plan robot behaviour using these models. However, the time required to perform planning can make this intractable for execution.

Finding policies that directly map from the current state to the next applicable action eliminates the need for planning. While Deep-Reinforcement Learning (D-RL) has shown promising results, for example those by OpenAI et al. (2018), D-RL training can be tedious and resource demanding. Plappert et al. (2017) report problems on the *HalfCheetah* environment where the algorithms converge to a local optimum corresponding to the cheetah wiggling on its back. They alleviated this problem by a different exploration scheme.

In preliminary experiments (not included in this paper) we found similar problems: D-RL algorithms were not able to learn a pushing task with a simulated 7-DoF robot arm. The algorithms we used were Deep Deterministic Policy Gradient (DDPG) Lillicrap et al. (2015) and Proximal Policy Gradient (PPO) Schulman et al. (2017) (from OpenAI Baselines by Dhariwal et al. (2017)).

The algorithms were also not reaching relevant parts of the state space. Consequently, and in line with the findings of Plappert et al. (2017) we assume that part of the problem of failing to learn good policies is related to insufficient exploration. To remedy this problem, one might increase search time while keeping exploration noise high, or use more principled exploration. While increasing search time will in the limit also yield acceptable solutions, directed exploration appears more promising to find good solutions more reliably and in less time.

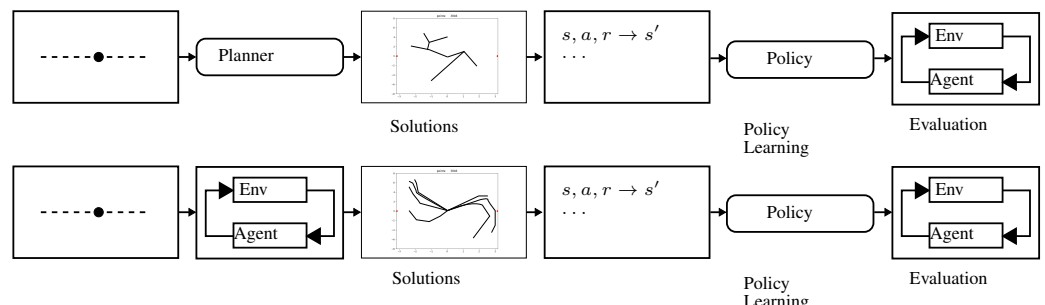

Figure 1: (top) The planner (LQR-RRT)(Perez et al., 2012) generates interactions — $(s, a, r, s')$ tuples — which are analyzed for state space coverage and are used in a SAC (Haarnoja et al., 2018) policy search process - see Figure 4. The returns that this policy generates are evaluated and compared to those of directly trained agents. (bottom) The agent directly learns a policy. The interactions it generates in this process are collected. These interactions are analyzed for state space coverage and are also used to train a policy. The return generated by these indirectly learned policies is also included in the comparison (Figure 4).

We thus focus on the latter approach, as covering a *more diverse area* of the state space increases the chances of finding an optimal solution, and moving away from random or exhaustive search reduces the number of samples required to learn a good policy.

Model-based methods can use their models of the task in an efficient way to plan over multiple steps and explore the state space in a more directed way. Given an accurate model, optimal policies can be produced without interacting with the world and thus with fewer samples (Hester & Stone, 2012). In particular, Rapidly Exploring Random Tree (RRT) are planning methods that focus on maximizing state-space exploration.

We propose to take advantage of the benefits the aforementioned planning methods provide while tackling the problem of planning time by synthesizing the planning results into a policy. This essentially makes the proposed method a model-based method (Sutton & Barto, 2018). We will refer to this method as *Planning for Policy Search (PPS)*. This is of particular interest in domain-randomized training, where simulation models are always available, to increase the data efficiency of exploration.

Here we investigate a preliminary version of this method that combines planning and policy search but does not perform randomizations yet. In particular, we investigate the following questions:

Q1 How do the data generated by RRT compare to those from D-RL methods? Do they cover a larger area of the state space? Do the reward distributions differ?

Q2 Are PPS methods less susceptible to local optima than D-RL methods?

Q3 Can the data collected by PPS be reused more easily?

The experimental setup used to investigate these questions is described in Figure 1 . In a simulated environment, the planner and reinforcement learning agent are run – each separately – to generate environment interactions. In the case of the reinforcement learning agent a policy is learned, and its return is evaluated (Q2, Sec. 4.2 ). In both cases, the collected data are stored as a dataset. In a second step, these datasets are analyzed with respect to their state-space coverage (Q1, Sec. 4.1 ). Then the datasets are used to train an RL agent in an off-policy fashion. The returns of this agent's policy are again evaluated (Q1, Sec. 4.1 ). In a further experiment an agent is trained partially from these datasets and partially from experience it generates (Q3, Sec. 4.3 ).

## 2 RELATED WORK

Using physically-based simulations for learning is limited by the necessity to approximate physical phenomena, causing discrepancies between simulated and real world results. This difference is called the *reality gap* and is a well-known problem in various fields of robotics. An important approach to cross the gap is *Domain randomization* (Tobin et al., 2017; Sadeghi & Levine, 2017;

James et al., 2017): instead of one simulated environment, learning is done using a distribution of models with varying properties – such as for example mass, friction, shape, position, force/torque noise, etc. The idea is to make the behavior policies learned by the reinforcement learning process more robust to the differences within this distribution, thereby increasing robustness against the difference between the training distribution and the target domain, i.e. against the reality gap.

The work from OpenAI has shown a successful use of domain randomization for learning in-hand-manipulation, however the number of required training steps is raised by a factor of 33 (OpenAI et al., 2018) when domain randomizations are introduced. This increases the number of training steps to the magnitude of about $3.9 \cdot 10^{10}$ from a magnitude of $1.2 \cdot 10^{9}$ – classical deep reinforcement learning approaches typically require $10^5$ to $10^9$ iterations of simulation steps – many algorithms are being tested on $10^6$ timesteps, depending on the environment. The required amount of training data can make this method prohibitively expensive and typically the availability of a simulation model is not exploited.

Improving the efficiency of domain randomization is an active topic of research, for example by using adversarial randomizations (Mandlekar et al., 2017) or limiting the training to stop before overfitting to idiosyncrasies of the simulation (Muratore et al., 2018). There are also reinforcement learning methods that are more sample-efficient such as guided policy search (Levine & Koltun, 2013) which is a model-based deep reinforcement learning method. In Guided Policy Search (GPS), rollouts from a deep neural network controller are optimised by an optimal control method such as the iterative Linear Quadratic Regulator (iLQR) (Todorov & Weiwei Li, 2005; Tassa et al., 2012) method. However, guided policy search is, depending on the task, usually initialized from demonstrations since the exploration capabilities of the underlying optimization method (iLQR) are limited. Furthermore, the optimization method requires an applicable, engineered cost function which is able to guide the search procedure towards relevant solutions.

The benefits from combining a model-based method with model-free reinforcement learning has been highlighted in Renaudo et al. (2014). However their work focuses on discrete problems and the model-based method and the model-free algorithm are controlling the agent together whereas we address continuous RL problems where the planning method produces data for the policy learner.

Affine Quadratic Regulator (AQR)-RRT (Glassman & Tedrake, 2010) or LQR-RRT (Perez et al., 2012) are examples of RRT methods, which use a dynamics-based cost metric to guide the tree extension, making this methods able to deal with kinodynamic planning problems.

The problem of performance is also recognized in planning and work is being undertaken to make RRT faster, for example by Wolfslag et al. (2018).

## 3 METHOD

The evaluations are done on a simple, one-dimensional double integrator task where the goal is to move a point mass to a goal position. The environment is illustrated and described in more detail in Table 1 . The reason for this choice is that this environment is easy to visualize and problems present in a simple task are likely to manifest as well in a more complex setting.

The environment contains two distinct goal locations. The agent receives a reward based on the distance to the goal points, where the reward is dominated by the distance to the closer one of the two goal locations. The placement of the goal locations is chosen such that the agent starts at a position where the gradient of the goal with the smaller reward is nonzero. This implies that simply maximizing the reward from the starting position will lead to a suboptimal policy and a smaller overall return.

The goals are located at position $-2.5$ and $6.0$, both at $0.0$ velocity. The placement is chosen such that a random action selection is more likely to stumble upon the lower reward solution ($-2.5$) rather than the higher reward solution ($6.0$). Figure 2 illustrates both the state space coverage in the case of uniformly-random actions without exploring starts and the reward received at each point in the state space. Although simple, this task is still relevant for robotics where the situation of a closer goal of less interest than a farther one is quite common (e.g. two charging batteries where the closest charges the robot more slowly than the furthest). Simply following the gradient of the reward might

Table 1: Description of the 1D double-integrator test environment: a point mass $M$ can be moved in a one-dimensional space $X = $ (position , velocity) by applying a continuous-valued force. Reward is received based on the distance to two possible goal locations $(G_1, G_2)$.

| Dynamics | | |
|---|---|---|
| $X = \begin{bmatrix} x \\ \dot{x} \end{bmatrix}$ | $\dot{x} = Ax + Bu$ |  |
| $A = \begin{bmatrix} 0 & 1 \\ 0 & 0 \end{bmatrix}$ | $B = \begin{bmatrix} 0 \\ 1 \end{bmatrix}$ | |

| Reward | | |
|---|---|---|
| $\max((1 - \tanh|X - G_1^*|), 2(1 - \tanh|X - G_2^*|))$ | $G_1 = \begin{bmatrix} -2.5 \\ 0.0 \end{bmatrix}$ | $G_2 = \begin{bmatrix} 6.0 \\ 0.0 \end{bmatrix}$ |

| Limits | | |
|---|---|---|
| $u \in [-1; 1]$ | $x \in [-10; 10]$ | $\dot{x} \in [-2.5; 2.5]$ |

also lead the robot to get stuck at obstacles and thus prevent it from successfully completing the task.

To have a broader baseline we included an *exploring-starts* variant of the environment where the initial state of the system is sampled uniformly from the state space. This is a sound algorithmic variant and easy to implement in this toy environment. However, in more complicated settings, such as a robot arm performing a pushing task, it may bring the agent into unreachable (disconnected) parts of the state space. Such unreachable parts could, for example, be locations that the robot cannot reach from its initial position, or position-velocity configurations that would be damaging to the robot. Furthermore, this would also imply randomizing the state of the objects the robot interacts with, which requires additional engineering effort. This is contrary to what we want to achieve by learning – which is why we assume that in many applications exploring starts are undesirable or impractical.

Unless otherwise noted, the algorithms are run for $10^5$ environment steps; the D-RL algorithms use 100-step episodes.

### 3.1 PPS & PLANNER

The PPS implementation we present here consists of an RRT-based planner to generate data and the SAC method to learn policies from that data and perform additional fine-tuning.

The planning method derives from the implementation of LQR-RRT by Perez et al. (2012). An RRT method consists of three components: *a)* a sampling method that decides where tree extensions should be directed to, *b)* a distance metric that estimates the cost of going from points in the tree to a new target point, and *c)* a local steering method, to reach from a given point to a target point in the state space.

Following the algorithmic description of LQR-RRT, we use an LQR-based distance metric and uniform sampling of the target locations but use a quadratic programming-based solver for finite-horizon steering between tree points and the target point. We only use the RRT not the RRT* variant. That is, we do not reconnect trajectories to find shorter paths – this is left to the finetuning.

### 3.2 BASELINE ALGORITHMS

We compare the performance to state-of-the-art D-RL algorithms, in particular PPO (Schulman et al., 2017; Hill et al., 2018), DDPG (Lillicrap et al., 2015) and SAC (Haarnoja et al., 2018).

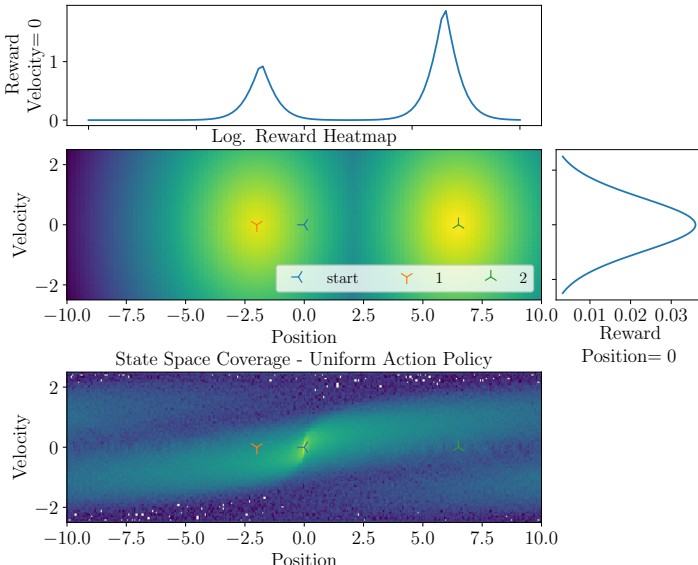

Figure 2: (Top) In the non-exploring-starts 1D double-integrator, the agent starts at position 0 with velocity = 0. The reward is based on the distance of the agent to the goal positions 1 and 2. Exactly at position 1 (resp. 2) the agent will receive a reward of 1 (resp. 2) at each step. The plots above and on the right of the heatmap show the reward distribution at velocity= 0 and at position= 0, respectively. (Bottom) When uniformly-sampled actions are taken in the 1D double integrator, the depicted state occupancy emerges. This shows which states the agent will stumble upon more easily and which states are harder to reach.

DDPG is an off-policy method that learns a deterministic policy using an actor-critic approach. Exploration is done by using the deterministic policy and adding exploration noise to the selected actions. In contrast, PPO works on-policy, uses a stochastic policy, is a policy-gradient method, and is related to Trust-Region Policy Optimization (TRPO) by Schulman et al. (2015). It tries to limit abrupt changes to the policy in order to keep generating reasonable data in the policy rollouts. PPO exploration works by sampling actions from its stochastic policy. Finally, SAC is an off-policy method, uses a stochastic policy and an actor-critic approach. Similarly to PPO, it explores by sampling actions from the stochastic policy. It adds an entropy term to the value-function loss to encourage more exploratory behaviour of the policy, that is, high entropy in the action selection is encouraged. Sec. 4.1 and Sec. 4.2 show how this affects the results.

We use the implementations provided by Hill et al. (2018) which are a tuned and improved version of the algorithms provided by Dhariwal et al. (2017). We use the default hyperparameters to investigate whether the algorithms are stable with respect to their hyperparameters. This is important, since either we view hyperparameter search as part of the policy search, or we require the algorithm to be robust to hyperparameter settings over a wide range of environments.

### 3.3 Q1 COMPARING DATA GENERATION

To compare the exploration, we collect the data the agents see during their learning phase. This data is then analysed for state-space coverage. The coverage is calculated as the percentage of non-empty bins. For simplicity we use uniformly-shaped bins. The number of bins is equal along each state-space dimension and is set to $\sqrt{10^5/5}$, i.e. such that, in the uniform case, we expect five data points in each bin on average. We calculate the coverage over time during the learning progress of the agent.

To evaluate the reward distributions of the algorithms, the data, $(s, a, r, s')$ tuples from 11 independent runs, are combined to form one dataset for each algorithm. We look at the distribution of the $r$ values of this dataset.

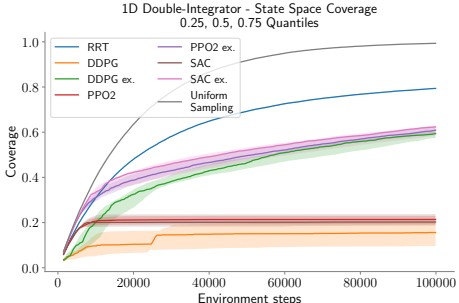 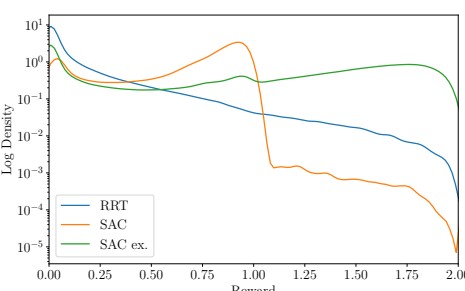

Figure 3: (Left) Coverage of the state space. (Right) The datasets generated by the methods SAC, SAC ex, and RRT consist of $(s, a, r, s')$ tuples. This plot shows how often what reward is achieved, or more precisely the log kernel density of the empirical distribution of rewards.

### 3.4 Q2 SUSCEPTIBILITY TO LOCAL OPTIMA

Reinforcement learning agents collect experience and use that experience to learn a policy, either implicitly in on-policy algorithms such as PPO, or explicitly in algorithms such as DDPG or SAC which use a *replay buffer*. Thus, the exploration process should reach regions in the state space relevant for the task so that it can learn a well-performing policy. If it cannot reach high-reward areas of the state space, the learned policy will also not move the agent to these regions and therefore the achieved return will be lower.

We perform training runs with PPO, DDPG, SAC and our PPS method. After the agents have learned, we use their policies to generate evaluation returns, which we analyze to compare their performance. This is done on 11 independent learning runs.

While the PPO, DDPG, SAC agents learn directly on the environment, our PPS agent uses the RRT planner to generate data. The generated data is stored in an SAC replay buffer. The replay buffer is fixed – no experience is added, no experience is removed. As a baseline this experiment is also performed for data generated by an SAC agent and an SAC agent on an exploring-starts variant. The data of both is also used in fixed SAC replay buffers to learn policies.

### 3.5 Q3 REUSING THE COLLECTED DATA

Since our PPS agent uses RRT with uniform sampling for tree extension to generate data, the state-space coverage is independent of the reward. It is therefore interesting to investigate whether this data can be reused more easily.

Similar to the previous experiment, a SAC agent is initialized with a random-weight-policy, but instead of an empty replay buffer, its buffer is preloaded with 50000 data samples created by one of the methods RRT, SAC, SAC ex. These samples are randomly shuffled to remove the order of their temporal acquisition. The agent then continues to acquire new environment interaction samples, which gradually replace the prefilled data in a First-In-First-Out (FIFO) fashion, while the policy is gradually updated on the new buffer. The agent is evaluated for another 50000 steps, however, the task is changed by disabling the reward at position 1 (resp. 2). As the buffer contains data about the previous task, only part of the previous dataset is useful whereas many samples are now misleading. If the agent has explored more during the previous task, we expect that its knowledge is more relevant to this second task and it should perform better than an agent that uses data generated in a more exploitation-focused way.

## 4 RESULTS

### 4.1 Q1 COMPARING DATA GENERATION

**How do D-RL methods compare in terms of exploration to a directed exploration approach, such as RRT? Can we cover a larger area of the state space?** Figure 3 shows the coverage

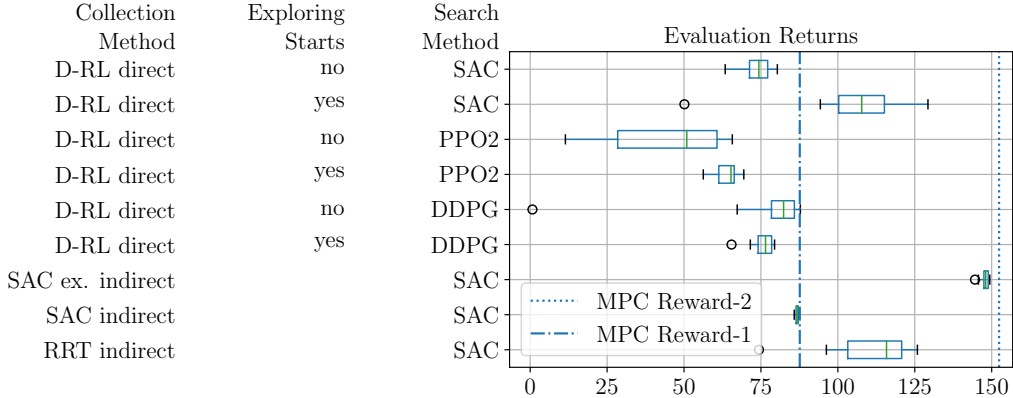

Figure 4: Box plot of the return distributions, generated from 11 independent runs where each run consists of the mean of 10 evaluation runs. The evaluation runs are performed at the end of the training process, equally spaced 10 episodes apart. The policies are trained directly or indirectly from interaction data (as described in Figure 1 ).

of the visited state space. Note how the RRT algorithm keeps increasing the state-space coverage while the D-RL agents level out. The exploring-starts agents (DDPG, PPO, SAC) follow the uniform sampling curve quite closely in the beginning, even exceeding the exploration of RRT, before the RRT method surpasses their coverage. Note how the RRT method surpasses the exploration of the non-exploring-starts methods from the very beginning. The agents trained with exploring starts (PPO, SAC, DDPG) level out at approximately the same coverage. A possible explanation is that by the exploring starts the agent starts from a random initial state but approaches the same favored goal. Thus the increase in coverage is mostly due to the exploring starts rather than the exploration mechanism of the agent. From this we conclude that the exploration capabilities built into methods such as DDPG, PPO and SAC are insufficient and that they mostly depend on the environment (i.e. exploring starts) to generate sufficiently-diverse data.

**Do the reward distributions differ?** Figure 3 shows how the rewards are distributed in the datasets collected by RRT, DDPG and SAC ex. respectively. For each method, the union of 11 runs is taken and the probability of achieving reward $r$ is calculated. The logarithm of the kernel density estimate of distribution of the rewards $r$ is depicted.

SAC favors the goal 1 position and as such most of the probability mass is concentrated around that reward. SAC ex. favors the second goal and consequently has a higher probability mass on the second goal. RRT collects data independently of the reward, which results in more samples around the higher reward goal (2) than SAC, but less than SAC ex. As such, RRT is more directed in exploring and consequently beats SAC in reaching the second goal. Both SAC and SAC ex. show a peak around their respective favored goal location while the data generated by RRT is independent of the reward and thus shows no such peaks. This also hints at the higher generality of the data generated by RRT which could be reused to achieve different goals, but also shows that very little data is generated around the regions of interest in this task.

## 4.2 Q2 SUSCEPTIBILITY TO LOCAL OPTIMA

**Do state-of-the-art D-RL methods get stuck in local optima?** Figure 4 depicts boxplots of the evaluation returns achieved by the D-RL algorithms after training for $10^5$ environment steps. Note how DDPG achieves higher rewards without exploring starts, and PPO appears to profit from exploring starts. SAC appears to profit from exploring starts, while otherwise achieving returns in a similar range to DDPG; in some cases it is able to achieve higher returns than both PPO and DDPG.

The figure also contains the evaluation returns of our PPS method, indicated by "RRT indirect SAC" and the direct baseline comparisons where data is generated by SAC and SAC ex., respectively, and is used indirectly to train an SAC policy.

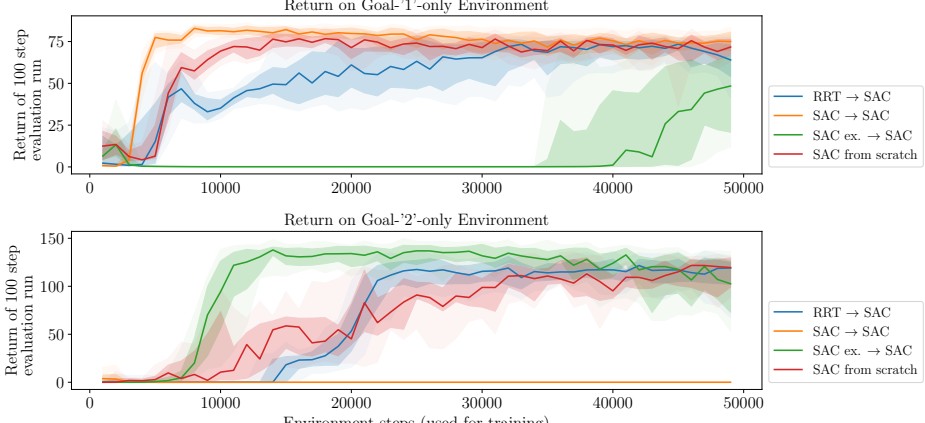

Figure 5: Evaluation returns of training a policy on a prefilled (with 50000 samples) replay buffer of size 50000 and training for 50000 steps. The highlighted area is the $25\%$ to $75\%$ range and the lightly-shaded area the $10\% - 90\%$ range.

The results show improved performance when training on the fixed replay buffer. They also show performance superior to the policy indirectly trained on SAC data as well as the directly-trained SAC policy. The policies trained on the RRT data even achieve performance comparable to the directly-trained SAC policy with exploring starts.

We use tanh activations in the indirectly-trained policies because we found them to produce results with smaller variance and to perform more robustly.

### 4.3 Q3 REUSING THE COLLECTED DATA

**Can the data collected by PPS be reused more easily?** In this experiment we train an SAC agent partially indirectly from a prefilled replay buffer but then continue with regular training, thereby phasing out the prefilled data. Figure 5 shows the evolution of the evaluation return distribution.

The data used for prefilling is generated on the two-goals enviroment, while the adaptation and evaluation is done on environments where one of the two goal locations is disabled. Therefore, only part of the prefilled data is accurate. In both cases the PPS method, denoted by RRT→SAC, is able to learn good policies, while the SAC→SAC agent has superior performance on the goal-"1"-only environment; it completely fails on the goal-"2"-only environment. The converse happens for the SAC ex.→SAC agent. It is interesting to note that part of the reused data is actually deceiving to the agent because it tries to get the agent to regions where reward can no longer be found. The learning-from-scratch agent is provided as a baseline. Since the modified environment is simpler – only one optimum – it achieves comparable results.

## 5 DISCUSSION

In this work, we highlighted that standard D-RL algorithms are not immune to getting stuck in suboptimal policies even in a toy problem with two local optima. The agent controlled by PPS explores a wider part of the state space than D-RL methods that focus on reward accumulation, even with exploring starts. The data gathered by RRT is not biased by reward accumulation and is thus more representative of the environment (goal 2 is farther away and thus incurs less reward).

We showed that the policy-learning agent trained on data from RRT performs better in the initial task than SAC but worse than SAC ex. However, on two variations of this task where only one source of reward is available, SAC ex. fails to adapt in half of the new tasks, whereas RRT achieves almost-optimal performance.

This method is thus relevant for robotics settings where the environment might dynamically change and some rewards might not be available after convergence of the robot policy (e.g. two sources of power are available in the environment at the beginning of the task and one becomes depleted during the robot's life).

This method also has the potential of speeding up domain-randomized training: By randomizing the model and using planning to quickly discover new policies, the method can focus the training on relevant parts of the state space and reduce the number of necessary samples. This will be evaluated in future work.

One limitation is that this evaluation is done on a simple task. It needs to be evaluated in more realistic settings where the state space is more complex and where variations of the task also alter environment dynamics.

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

## A   APPENDIX

