# OpenReview forum: "Improving Exploration of Deep Reinforcement Learning using Planning for Policy Search"
_ICLR.cc/2020/Conference — Reject_

### Official Review · AnonReviewer3 · 2019-10-17
**Official Blind Review #3**

**Rating:** 3

**Review:**

This paper suggested that conventional deep-reinforcement learning (D-RL) methods struggle to find global optima in toy problem when two local optima exist.  The authors proposed to tackle this problem using planning method (Rapidly Exploring Random Tree, RRT) to expand the search area. Since the collected data are not correlated with reward, it is more likely to find the global optima in toy problem with two local optima . As to the planning time problem,  they proposed to synthesize the planning results into a policy.

The experiments proved that the proposed method performs better in the aforementioned toy problem, and  has advantage in adapting dynamic environment. However, the authors failed to provide sufficient analyis and theoretical support for the proposed method, plus it did not address the weakness of the RRT method-the problem of planning time.

**Experience Assessment:**

I do not know much about this area.

**Review Assessment: Checking Correctness Of Derivations And Theory:**

I assessed the sensibility of the derivations and theory.

**Review Assessment: Checking Correctness Of Experiments:**

I carefully checked the experiments.

**Review Assessment: Thoroughness In Paper Reading:**

I read the paper thoroughly.

---

> ### Author Response · Authors · 2019-11-15
> **Comment**
>
> • Thanks for your review!
> • Our proposed method uses a planning method (in this implementation RRT) in the learning and data collection phase - which is then used to learn a policy. During execution the policy
> is used thereby eliminating the planner time from policy execution.
> • The time taken by the planner during the offline data collection phase is not evaluated in our paper yet - we will add that in a future extension.

---

### Official Review · AnonReviewer2 · 2019-10-23
**Official Blind Review #2**

**Rating:** 1

**Review:**

The paper aims to improve exploration in DRL through the use of planning. This is claimed to increase state space coverage in exploration and yield better final policies than methods not augmented with planner derived data.

The current landscape of DRL research is very broad, but RRT can only directly be applied in certain continuous domains with continuous action spaces. With learned embedding functions, RRT can be applied more broadly (see "Taking the Scenic Route: Automatic Exploration for Videogames" Zhan 2019). The leap from RRT-like motion planning to the general topic of "planning" for policy search is not well motivated explained with respect to the literature. Uses of Monte Carlo Tree Search (as in AlphaGo) seem obviously related here.

This reviewer moves to reject the paper primarily on the grounds of overinterpreting experimental results from a single, extremely simple example RL task. In a domain so small, we can't tease out the role of exploration, we aren't engaging with the "deep" of DRL, and we are only considering one specific kind of planning. The implicit claims of general improvement to exploration and improved downstream policies are not supported by the experimental results. At the same time, no theoretical argument is attempted that would make up for the very narrow nature of the experiments.

Questions for the authors:
- If HalfCheetah is used to motivate the work, and it is so easily available in the open source offerings from OpenAI, why isn't one (or many more) tasks of *at least* this complexity considered? MountainCar is one of the gym environments with a 2D phasespace compatible with the kinds of plots used in this paper.
- Could the authors taxonomize the landscape of planning and provide a specific argument for focusing on RRT? (RRT is a fun algorithm, but how will you draw the attention of other researchers who are currently focused on Atari games?)

**Experience Assessment:**

I have published one or two papers in this area.

**Review Assessment: Checking Correctness Of Derivations And Theory:**

N/A

**Review Assessment: Checking Correctness Of Experiments:**

I assessed the sensibility of the experiments.

**Review Assessment: Thoroughness In Paper Reading:**

I read the paper at least twice and used my best judgement in assessing the paper.

---

> ### Author Response · Authors · 2019-11-15
> **Comments on Questions**
>
> • Thank you for your review!
> • We eventually want to apply our method on robotics tasks and therefore we focus on continous state/continuous action spaces.
> • The paper by Zhan et al. ’19 (“Taking the scenic route: Automatic exploration for video games”) shows an interesting idea to extend the use of RRT even to domains like the Atari games: they use features of a neural network as a low-dimensional continuous embedding of an (Atari/similar) game state (i.e. the image). They use a simplified version of RRT that samples a target point, restores the closest state stored in the tree and tries to reach that target
> point. However usually RRT uses a local steering method to reach that target point – while Zhan et al. use a (random) action sequence irrespective of the target point.
> Since we want to target to robotics tasks, where action and feature spaces are inherently continuous and discretization becomes infeasible, we need to be able to deal with such action spaces. Moreover and related to the next comment, random steering will often not be beneficial: either it cannot be done at all, or it is too inefficient.
> • MCTS: Since we want to eventually apply our method on robotic tasks, we are focusing on continuous domains, we focus on planning methods that are able to deal with such domains and therefore sampling based planners. AlphaGo is an impressive showcase of an extended version of MCTS - however MCTS needs extensions to be applicable in continuous domains, such as
> for example (but not limited to) HOOT (Mansley et al.,“Sample-Based Planning for Continuous Action Markov Decision Processes”), to be applicable to continuous action domains.
> A second aspect is that MCTS typically does not use the information often available in these continuous domains: the locally linearised dynamics - which the local steering method of RRT exploits.
> We therefore chose RRT as a reasonably effective, yet reasonably simple to implement planning method - although we do not foresee any reason why other planning methods would not work as long as they are applicable to kino-dynamic domains, and produce environment interactions.
> • We will add more tasks in the next extension.

---

### Official Review · AnonReviewer1 · 2019-10-23
**Official Blind Review #1**

**Rating:** 1

**Review:**

The paper is mostly easy to read and I enjoyed reading it. The authors address an important issue of exploration in reinforcement learning and the used of a model-based planner is certainly a promising direction. However, I do have a number of concerns.

1. On Q1. I think the key question here is this -- should state-space coverage be the only measure for effective exploration? The classical dilemma of explore-or-exploit in reinforcement learning is relevant here. From Figure 3, it seems that RRT tends to explore uniformly rather than "intelligently". For problems where there is absolutely no information guiding the exploration process this might be desirable, but then the search complexity will suffer from the curse of dimensionality and there is no evidence in this work that this is a good strategy. Perhaps switching from RRT to RRT* helps but the authors chose not to do it.

2. On Q2. Perhaps I missed something here but other than special cases (e.g. convex problems) almost all gradient-based algorithms suffer from local optimality. I am not sure Q2 is a good question to ask here.

3. On Q3. It seems that SAC from scratch is the best-performing approach here. This particular setting is hardly convincing in motivating the re-use of examples across tasks.

The above concerns, plus the fact that only one particularly simple task is being investigated here, prevent me from recommending acceptance.


**Experience Assessment:**

I have published in this field for several years.

**Review Assessment: Checking Correctness Of Derivations And Theory:**

N/A

**Review Assessment: Checking Correctness Of Experiments:**

I carefully checked the experiments.

**Review Assessment: Thoroughness In Paper Reading:**

I read the paper thoroughly.

---

> ### Author Response · Authors · 2019-11-15
> **Comment**
>
> • Thank you for your review!
> • @1.) Eventually, probably the most interesting final metric is task success and therefore achieved return – but that strongly depends on the task.
>
> Without assumptions on how the reward is structured (with respect to the state space), it is not possible to exclude portions of the state space and without excluding regions of the state space it is not possible to explore more intelligently.
>
> In dynamical systems, the distance between two state space vectors cannot be measured by simply taking the euclidean distance – this is due to the dynamics (under-actuation, obstacles). A simple example is a pendulum swing-up, the mountain car example or a robot in a maze. While the target location (in the state space) might be close in terms of euclidean distance, it may not be possible to reach that point (not enough torque and force, or a blocking wall). As such it is hard to define guiding assumptions for the exploration. As such uniform exploration of the state space appears to be a crude but reasonable approach.
>
> • @1.) Curse of dimensionality: The method will suffer from the curse of dimensionality, however, this is also true for other methods - probably ways to deal with this problem are a) to reduce the high dimensional problem to a lower dimensional one, or b) to use heuristics and solve it only approximately.
>
> One benefit of RRT and the local steering method is that even in high dimensional spaces the tree will span the state space coarsely if the dynamics allow that.
>
> • @1.) RRT/RRT*: The difference between RRT and RRT* is that RRT* finds optimal paths from the initial point to the target points in the state space, while the paths found by RRT are not optimal (i.e. not the shortest paths) - however, RRT* requires additional environment steps to perform this optimization - whereas we mostly want to use RRT to find ways to reach large areas of the state space and optimize around the most promising regions.
> This is also visible in Figure 5, where training is done from 50k RRT steps (full exploration) and then slowly replaced by samples from SAC - thereby fading from pure exploration to exploitation. Although unfortunately we did not highlight this aspect well.
>
> • @2.) While gradient-based algorithms suffer from local optimality, it does not feature prominently in D-RL research. And given the large amount of excitement around D-RL and the impressive success (e.g. the OpenAI work using the Shadow Dexterous Hand, although this was achieved with great effort - thirteen thousand years of experience) - we felt it beneficial to show that this is a problem that actually happens in practise and therefore is relevant.
> • We will include more experiments in a future extension of this paper.

---

### Decision · Program_Chairs · 2019-12-19

**Decision:**

Reject

**Comment:**

The paper is about exploration in deep reinforcement learning. The reviewers agree that this is an interesting and important topic, but the authors provide only a slim analysis and theoretical support for the proposed methods. Furthermore, the authors are encouraged to evaluate the proposed method on more than a single benchmark problem.